# Mixture-of-Experts with Expert Choice Routing

Yanqi Zhou, Tao Lei, Hanxiao Liu, Nan Du, Yanping Huang, Vincent Y. Zhao, Andrew Dai, Zhifeng Chen, Quoc Le, and James Laudon

Google, Mountain View, CA, USA
{yanqiz, taole, hanxiaol, dunan, huangyp, vzhao, adai, zhifengc, qvl, jlaudon}@google.com

## Abstract

Sparsely-activated Mixture-of-experts (MoE) models allow the number of parameters to greatly increase while keeping the amount of computation for a given token or a given sample unchanged. However, a poor expert routing strategy can cause certain experts to be under-trained, leading to an expert being under or over-specialized. Prior work allocates a fixed number of experts to each token using a top-$k$ function regardless of the relative importance of different tokens. To address this, we propose a heterogeneous mixture-of-experts employing an expert choice method. Instead of letting tokens select the top-$k$ experts, we have experts selecting the top-$k$ tokens. As a result, each token can be routed to a variable number of experts and each expert can have a fixed bucket size. We systematically study pre-training speedups using the same computational resources of the Switch Transformer top-1 and GShard top-2 gating of prior work and find that our method improves training convergence time by more than $2\times$. For the same computational cost, our method demonstrates higher performance in fine-tuning 11 selected tasks in the GLUE and SuperGLUE benchmarks. For a smaller activation cost, our method outperforms the T5 dense model in 7 out of the 11 tasks.

## 1 Introduction

Scaling up model capacity, dataset size, and training time has demonstrated huge success in enhancing the performance of computer vision architectures [4, 11, 13, 14] as well as neural language models [2, 20, 26, 27]. The final model quality has been found to have a power-law relationship with the amount of data, model size, and compute time [16, 20]. However, training efficiency, which is defined as the total amount of computation used to achieve superior model quality than the state of the art system [21], should receive greater attention as we increase our efforts towards green AI [29].

Sparsely gated mixture-of-experts [31] (MoE) provides an effective way to scale model capacity given a fixed computational cost, and has recently played an important role in increasing the training efficiency of large-scale language models [10, 21]. MoE operate by adopting a number of experts, each as a sub-network, and by activating only one or a few experts for each input token. A gating network must be chosen and optimized in order to route each token to the most suited expert(s). For example, recent work has implemented sparse routing via $k$-means clustering [12], linear assignment to maximize token-expert affinities [22], or hashing [8, 28]. Many of the prior work use a routing strategy concerning the *token choice*, where each token selects the best one or two experts.

We argue that the independent token choice of prior work often leads to an imbalanced load of experts, which causes training inefficiency and sub-optimal training of the model. In order to mitigate this

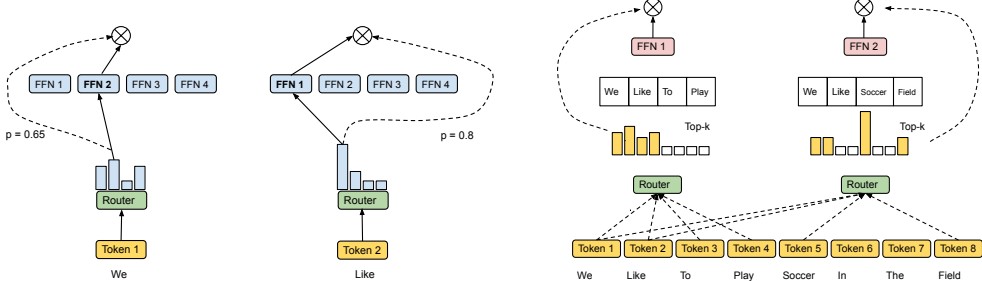

Figure 1: High-level Comparison Between Conventional MoE and expert choice MoE.

issue, previous sparsely gated networks introduce additional auxiliary losses as regularization to prevent too many tokens being routed to a single expert, but the effectiveness is still limited. Recent approaches [8, 22, 28] explore alternative strategies for routing, but they focus on pre-training only and do not demonstrate performance gain on downstream tasks. Moreover, none of the previous methods consider allocating a variable number of experts to each token based on importance, which can be beneficial.

We propose a very simple yet effective routing method we are calling *expert choice*. Unlike conventional MoE where tokens select one or two top-scoring experts, our method lets each *expert* pick the top-$k$ tokens. Our method guarantees perfect load balancing, allows a variable number of experts for each token, and achieves substantial gains in training efficiency and downstream performance as demonstrated in our experiments. Our major contributions include:

- We identify common pitfalls in conventional MoE such as load imbalance as described in Section 3.1. We then propose a heterogeneous, expert choice method to provide a fluid allocation of model parameters based on a learnt token-to-expert importance. This method intrinsically guarantees load balance without imposing an auxiliary loss.

- We show our method provides over $2\times$ faster training convergence in a 8B/64E (8 billion activated parameters, 64 experts) model, compared to the top-1 and top-2 gating counterparts in Switch Transformer [10] and GShard [21].

- We show our method demonstrates strong scaling when increasing the number of experts from 16 to 128, evaluated in training perplexity.

- We show our method demonstrates strong performance on downstream tasks selected from GLUE and SuperGLUE at all the evaluated scales. More specifically, our 8B/64E model outperforms a T5 11B dense model in 7 out of 11 tasks evaluated.

## 2   Related Work

**Scaling:** Various approaches have been proposed to scale up neural network capacity to improve performance. Recent works have successfully scaled models to billions of parameters via various forms of model parallelism [2, 21, 26, 27, 33]. Model parallelism [30] splits weights and tensors across multiple cores while pipeline parallelism [18, 24] splits different layers across devices with micro-batches pipelined to the different layers. To enable continued scaling of neural networks, improving model training and serving efficiency has become a critical research area.

**Conditional Computation:** Computation decisions can be made dynamically based on the input [23, 25]. Conditional computation has been proposed as a way to increase the capacity of a deep neural network without increasing the amount of computation, by activating certain parameters and computation on demand, on a per-example or per-token basis [3]. Conditional convolution layers [1] with task-specific gating has been used to combat catastrophic forgetting when a sequence of learning problems are optimized. The gating decisions may be binary or sparse and continuous, stochastic or deterministic.

**Mixture of Experts:** Sparsely-gated MoE [31] is the first model to demonstrate massive improvements in model capacity, training time, or model quality with gating. Switch Transformer [10] simplifies the gating by selecting only the top expert per token using a softmax over the hidden state and demonstrates better scaling than previous work. All the prior work requires an auxiliary loss to explicitly encourage balancing. This loss term has to be carefully weighted to not overwhelm the primary loss. However, auxiliary loss does not guarantee balancing and a hard capacity factor has to be imposed. As a result, many tokens can still be unprocessed by the MoE layer. Hard MoE [12] with a single decoding layer can be efficiently trained to good effect on large scale hashtag prediction tasks. Base Layers [22] formulate a linear assignment that maximizes token-expert affinities while ensuring each expert receives an equal number of tokens. Hash layers [8, 28] devise hashing techniques on input tokens. However, the evaluations are limited to pre-training perplexity. THOR [37] randomly activates experts during training and inference and is trained with a consistency regularization loss. THOR has demonstrated strong performance on translation tasks. Different from these prior works, our method is a learnt method that enables heterogeneous MoE and effectively improves downstream fine-tuning performance.

## 3 Method

We first identify a few pitfalls in the routing method of conventional mixture-of-experts (MoE) models and then present our method using expert choice to tackle these problems.

### 3.1 Pitfalls of Token-Choice Routing

MoE can be computationally advantageous compared to a dense model, a routing strategy must be used to assign each token to the most-suited experts. Conventional MoE models employ *token-choice* routing which independently selects the top-$k$ experts for each token [10, 21, 31]. We argue that this strategy has a few pitfalls that lead to sub-optimal training.

**Load Imbalance:** Token-choice routing often lead to poor load balancing across experts. That is, some experts may be trained with most tokens, leaving the remaining experts under-utilized. Experts can be under specialized because a lot of model capacity in the under-utilized experts are wasted. On the other side, some tokens will not be processed, since over-utilized experts can only take a maximum number of tokens at each step in order to avoid running out of memory. Load imbalance can also hurt step latency, thus inference time, as the step latency can be determined by the most loaded expert. Previous methods add an auxiliary loss on load balancing to mitigate the issue. However, this auxiliary loss does not guarantee a balanced load, especially during the important early stages of training. Indeed, **we empirically observe that the over-capacity ratio can reach 20%–40% for some experts in token choice routing**, indicating that a significant portion of the tokens routed to these experts will be dropped.

**Under Specialization:** Each MoE layer uses a gating network to learn token-to-expert affinity. Ideally, the learnt gating network should produce the affinity such that similar or relevant tokens are routed to the same expert. A sub-optimal strategy can produce redundant experts and/or experts that are not sufficiently specialized. Under specialization may result by imposing an large auxiliary loss which favors more load balanced but less effective routing. Finding the right balance on the auxiliary loss to promote both load balancing and specialization is challenging for token-choice routing.

**Same Compute for Every Token:** Finally, in a token-choice strategy each token receives exactly $k$ experts and therefore occupies the same amount of compute. We hypothesize that this is not necessary nor desired. Instead, a MoE model should flexibly allocate its compute resource based on the complexity of the input. Motivated by the aforementioned observations, we next describe a simple yet effective method which produces load balanced assignments based on *expert choice*.

### 3.2 Heterogeneous MoE via Expert Choice

Different from conventional routing, an expert choice method independently selects top-$k$ tokens for each expert, where $k$ is a fixed expert capacity (i.e. the number of tokens each expert can take). Despite its simplicity, expert choice achieves perfect load balancing by design. It also enables more flexible allocation of model compute since tokens can be received by a variable number of experts.

In our experiments, we set $k$ as

$$k = \frac{n \times c}{e} \tag{1}$$

where $n$ is the total number of tokens in the input batch (such as batch size $\times$ sequence length), $c$ is the capacity factor, and $e$ is the number of experts. The capacity factor $c$ denotes on average how many experts are utilized by a token. Given input token representations $X \in \mathbb{R}^{n \times d}$ where $d$ is the model hidden dimension, our method produces a token-to-expert assignment denoted by three output matrices $I$, $G$ and $P$. The matrix $I$ is an index matrix where $I[i, j]$ specifies $j$-th selected token of the $i$-th expert. The gating matrix $G \in \mathbb{R}^{e \times k}$ denotes the weight of expert for the selected token, and $P \in \mathbb{R}^{e \times k \times n}$ refers to an one-hot version of $I$ that will be used to gather tokens for each expert. These matrices are computed using a gating function,

$$
\begin{aligned}
S &= \mathrm{Softmax}(X \cdot W_g), \quad S \in \mathbb{R}^{n \times e} \\
G, I &= \mathrm{TopK}(S^\top, k), P = \mathrm{Onehot}(I)
\end{aligned} \tag{2}
$$

where $S$ denotes the token-to-expert affinity scores, $W_g \in \mathbb{R}^{d \times e}$ denotes the expert embeddings, and $TopK()$ selects the k largest entries for each row of $S^\top$.

Similar to Switch Transformer [10] and GShard [21], we apply mixture of experts and the gating function in the dense feed-forward (FFN) layer, as it is the most computationally expensive part in a Transformer-based network. The input to the gated FFN, denoted by $X_{\mathrm{in}} \in \mathbb{R}^{e \times k \times d}$, is produced using the permutation matrix $P$. Here $X_{\mathrm{in}}[i] \in \mathbb{R}^{k \times d}$ denotes the input of the $i$-th expert. Similarly, let $W_1$ and $W_2$ denote the parameters of gated FFN in which $W_1[i]$ and $W_2[i] \in \mathbb{R}^{d \times d'}$ denote the parameter matrices of the $i$-th expert. We compute the output of each expert $X_e[i]$ as follows,

$$
\begin{aligned}
X_{in} &= P \cdot X \\
\forall i : \quad X_e[i] &= \mathrm{GeLU}(X_{in}[i] \cdot W_1[i]) \cdot W_2[i]^\top
\end{aligned} \tag{3}
$$

We omit the bias terms here for brevity. The finally output of the gated FFN layer $X_{\mathrm{out}} \in \mathbb{R}^{n \times d}$ can be obtained given $X_e$, the permutation and gating matrices $P$ and $G$,

$$X_{\mathrm{out}}[l, d] = \sum_{i,j} P[i, j, l]\, G[i, j]\, X_e[i, j, d] \tag{4}$$

Both $X_e$ and $X_{\mathrm{out}}$ can be efficiently computed using Einstein summation (einsum) operations.

### 3.3 Expert Choice with Additional Constraint

We also consider regularizing our expert choice routing by limiting the maximum number of experts for each token. We are interested in whether adding this constraint improves pre-training and fine-tuning results. More importantly, it helps analyzing to what degree using a variable number of experts per token affects the model performance.

Let $A \in \mathbb{R}^{e \times n}$ be a positive matrix where $A[i, j]$ represents whether the i-th expert selects j-th token. We solve the following entropy-regularized linear programming problem

$$
\max_A \; \langle S^\top, A \rangle + \lambda H(A)
$$
$$
\text{s.t.} \quad \forall i : \sum_{j'} A[i, j'] = k; \;\; \forall j : \sum_{i'} A[i', j] \leq b; \;\; \forall i, j : 0 \leq A[i, j] \leq 1
$$

where $< S^\top, A >$ denotes the inner product, $H(A)$ is the sum of element-wise entropy[1], and $b > 0$ is an integer that upper bounds the selection for each token. Adding a small entropy term gives a near-integer solution while enabling a fast iterative solver we can run on TPUs. Specifically, the solution space is the intersection of three convex sets each satisfying one of the linear constraints. We use Dykstra's algorithm [9] that alternatively projects the intermediate solution onto one of the convex sets.[2] After $A$ is computed, the routing indices $I$ is selected using $TopK(A, k)$ instead.

---

[1] $H(A) = \sum_{ij} -A[i, j] \log A[i, j]$
[2] We use $\lambda = 0.001$ and a maximum of 100 iterations.

| Model | Type | $n_{\text{params}}$ | $n_{\text{act-params}}$ | $L$ | $M$ | $H$ | $n_{\text{heads}}$ | $d_{\text{head}}$ | $E$ |
|-------|------|------|------|------|------|------|------|------|------|
| 0.1B | Dense | 130M | 130M | | | | | | - |
| 0.1B/16E | MoE | 548M | 145M | | | | | | 16 |
| 0.1B/32E | MoE | 1.0B | 145M | 12 | 768 | 3,072 | 12 | 64 | 32 |
| 0.1B/64E | MoE | 1.9B | 145M | | | | | | 64 |
| 0.1B/128E | MoE | 3.7B | 145M | | | | | | 128 |
| 8B | Dense | 8.7B | 8.7B | 32 | 4,096 | 16,384 | 32 | 128 | - |
| 8B/64E | MoE | 143B | 9.8B | | | | | | 64 |

Table 1: Sizes and architectures of both MoE and dense models that were trained in our experiments. Models are grouped by the number of activated parameters per token. All trained models share the same learning hyperparameters described in Section 4.1.

## 3.4 Model Architecture

At the high level, we adopt the idea of sparsely activated Mixture-of-Experts (MoE) [31]. We use a Transformer architecture and replace the feed-forward component of every other Transformer layer with a MoE layer, following recent practice [10, 21]. Interleaving regular Transformer layers and MoE layers empirically improves model performance and training efficiency, probably because forcing some shared components in between MoE layers can mitigate the negative effects of skipping tokens. Several additional modifications adopted in recent work have been applied in our experiments. For example, we replace the standard positional embedding with per-layer relative positional bias [5]. In the non-MoE feed-forward sub-layers (only every other layers are MoE layers), we replace the first linear projection and the activation function with the Gated Linear Unit [6], which computes the component-wise product of two linear transformation of the input, followed by a Gaussian Error Linear Unit [15] activation function.

As described earlier, each MoE layer consists of a group of independent feed-forward networks as denoted as "experts". The gating function in Eq. (2) uses a softmax activation function to model a probability distribution over these experts. This distribution denotes the preference over experts of each incoming token, which is computed similarly in a conventional gating network [10, 21, 31]. During training, each MoE layer's learnable gating network described in Eq. (2) is trained to use the input to activate the best subset of experts using a top-$k$ function along the token dimension. An "shuffle" stage and an "unshuffle" stage are inserted to the MoE layer, where the first stage gathers the tokens to their designated experts while the second stage permutes the tokens back to their original order in the input batch. This step is formulated in Eq. (3) and Eq. (4).

Similar to conventional MoE method, there are more parameters in the MoE layer. However, the activated model size per token can be comparable to a dense layer because during training or inference, only a limited subset of experts is activated for any given token. For instance, Switch Transformer [10] has only one activated expert while GShard [21] uses two experts per token. In our method, the number of activated experts can vary for each token but the overall computation is kept the same as the baseline architectures by fixing the capacity factor $c$ in Eq. (1). Unless otherwise specified, we set $c = 2$ such that our method can be directly compared to the top-2 token-choice gating in GShard.

We train several variants of our architecture at the 100M scale (i.e. 100M expert size) by increasing the number of experts to understand the scaling effects of our method. We also train a 8B scale MoE model. The large MoE model is partitioned with a 2D sharding algorithm as presented in GSPMD [36], which fully exploits the 2D topology of the TPU cluster [19]. Across different scales and setups, our method outperforms related work and demonstrates strong downstream task performance on selected tasks in GLUE and SuperGLUE.

## 4 Experiments

### 4.1 Setup

Table 1 summarizes the hyperparameter settings of different MoE models. As a reference point, we also include the respective dense model configurations with comparable numbers of activated parameters per-token during inference. To study of the effect of scaling the number of experts, we

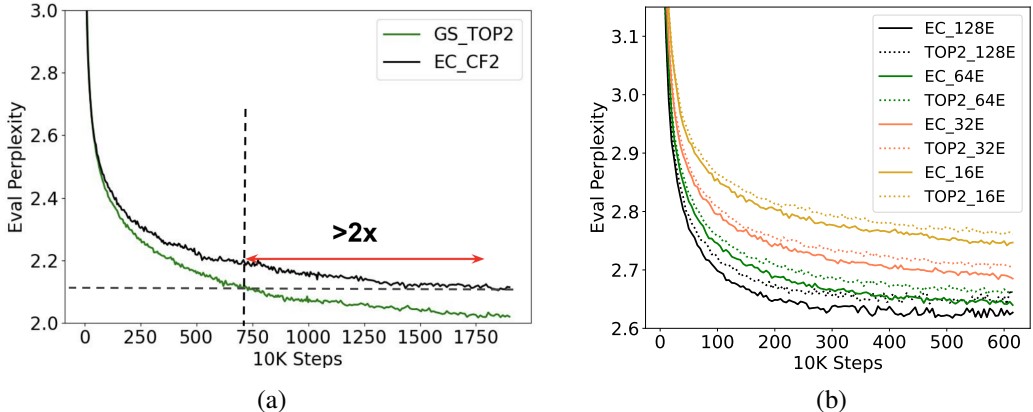

Figure 2: (a) Training convergence is more than 2x faster using our method compared to GShard top-2 gating. (b) Training perplexity scales strongly with the number of experts while keeping the expert size fixed. EC consistently outperforms GShard top-2 gating.

studied varying the number of experts but fixing the per expert size to 100M parameters. For example, 0.1B/64E represents the architecture of an approximately 100M parameter dense model with every other layer replaced by a 64-expert MoE layer. The MoE model degenerates into a dense transformer architecture when each MoE layer only has one expert. While $n_{params}$ is the total number of trainable parameters, $n_{act-params}$ represents the number of activated parameters per token. $L$ is the total number of Transformer layers, $M$ is the model dimension, $H$ is the hidden dimension after the projection in each transformer layer, $n_{heads}$ is the number of attention heads, and $d_{head}$ is the hidden dimension of each attention head.

**Dataset:** We use the high-quality dataset from GLaM [7] of 1.6 trillion tokens that are representative of a wide range of natural language use cases. An in-house classifier is trained to classify between a collection of curated text and other webpages and estimate the content quality of a webpage. A high-quality filtered subset of webpages are combined with books, Wikipedia pages, conversations, forums, and news to create the final dataset. The data and mixture weights can be found in Table 3 in the GLaM paper.

**Model Training:** Our model training follows the setups of GLaM [7] where a maximum sequence length of 1024 tokens is adopted. We use an Adafactor optimizer [32] with first-moment decay $\beta_1 = 0$ and second-moment decay $\beta_2 = 0.99$. We keep the learning rate constant for the first 10K training steps, and then decay it with an inverse square root schedule. Unlike most related works, we do not impose any auxiliary loss for load balance, such as described in Switch Transformer [10] and GShard [21]. We use the SentencePiece subword tokenizer with a vocabulary of size of 256K. The largest model (8B/64E) is trained on 512 TPU V4 chips. We use a dropout rate of 0 during training as the number of tokens in the training data corpus is much greater than the total number of tokens during training.

**Model Evaluation:** We mainly focus on evaluating the finetuning performance on the 11 selected tasks from GLUE and SuperGLUE benchmarks [34, 35].

## 4.2 Training Efficiency

We first study training efficiency and convergence. We use expert choice with a capacity factor of 2 (EC-CF2) to match the activated model size and computational cost on a per token basis in GShard top-2 gating and run both for a fixed number of steps. The results are shown in Fig. 2 (a). Comparing to GShard top-2 gating, which showed stronger performance in both perplexity in the evaluation dataset and fine-tuning on downstream tasks compared to Switch Transformer top-1 gating, EC-CF2 converges more than 2x faster during training. More specifically, EC-CF2 reaches the same perplexity as GShard top-2 in less than half the steps, and with each GShard top-2 step being 20% slower than our method. As explained in Section 3.1, the slower step time in top-2 gating is due to load imbalance

| | | | 100M/128E | | | 100M/64E | | |
|---|---|---|---|---|---|---|---|---|
| Name | Metric | Split | ST Top-1 | GS Top-2 | EC-CF2 | ST Top-1 | GS Top-2 | EC-CF2 |
| BoolQ | acc | dev | 77.4 | 76.5 | **76.9** | 73.2 | 77.5 | **79.7** |
| CB | acc | dev | 87.5 | 80.9 | **89.1** | 85.9 | 84.4 | **89.1** |
| CoLA | acc | dev | 78.9 | 84.0 | **86.7** | 64.1 | 85.2 | **88.3** |
| MNLI | acc | dev | 82.3 | 83.6 | **84.9** | 80.8 | 85.2 | **86.7** |
| MRPC | acc | dev | 82.6 | 81.0 | **83.1** | 81.3 | 81.3 | **84.4** |
| QNLI | acc | dev | **89.5** | 88.6 | 89.0 | 89.4 | 89.7 | **91.3** |
| QQP | acc | dev | **90.6** | 90.3 | 90.4 | 88.9 | 90.5 | **91.0** |
| RTE | acc | dev | 77.0 | **78.9** | 78.5 | 74.1 | 79.3 | **81.6** |
| SST2 | acc | dev | 92.0 | 94.5 | **94.6** | 91.8 | 95.1 | **95.1** |
| WiC | acc | dev | 67.8 | 65.5 | **68.1** | 64.4 | **67.8** | 65.6 |
| WNLI | acc | dev | 65.6 | **70.3** | 67.2 | 68.8 | 68.8 | **71.7** |
| | | | | | | | | |
| Avg | - | - | 81.0 | 81.3 | **82.6** | 78.4 | 82.2 | **84.0** |

| | | | 100M/32E | | | 8B/64E | | |
|---|---|---|---|---|---|---|---|---|
| Name | Metric | Split | ST Top-1 | GS Top-2 | EC-CF2 | ST Top-1 | GS Top-2 | EC-CF2 |
| BoolQ | acc | dev | 74.5 | 79.0 | **79.3** | 89.1 | **89.5** | 89.2 |
| CB | acc | dev | 80.6 | 81.3 | **92.2** | 93.8 | 96.7 | **100** |
| CoLA | acc | dev | 87.5 | 92.2 | **93.8** | 88.3 | 87.5 | **89.1** |
| MNLI | acc | dev | 83.1 | 87.8 | **88.0** | 90.7 | **91.4** | 91.1 |
| MRPC | acc | dev | 82.3 | **85.2** | 84.4 | 89.3 | **91.7** | 90.6 |
| QNLI | acc | dev | 91.6 | 91.9 | **92.5** | 94.5 | 94.9 | **95.0** |
| QQP | acc | dev | 90.1 | 91.5 | **92.0** | 92.1 | 92.5 | **93.8** |
| RTE | acc | dev | 75.0 | **79.1** | 78.1 | 91.0 | 92.2 | **95.2** |
| SST2 | acc | dev | 93.3 | 94.4 | **95.4** | 97.1 | **98.0** | 97.7 |
| WiC | acc | dev | 62.5 | 65.9 | **69.8** | 74.5 | 76.4 | **83.8** |
| WNLI | acc | dev | 65.6 | 64.1 | **68.8** | 78.1 | 82.8 | **92.8** |
| | | | | | | | | |
| Avg | - | - | 80.6 | 83.5 | **85.0** | 88.9 | 90.3 | **92.6** |

Table 2: Expert choice with capacity factor of 2 (EC-CF2) outperforms Top-1 gating in Switch Transformer (ST) and top-2 gating in GShard (GS) on GLUE and SuperGLUE tasks. Note that with an expert size of 100M parameters, 100M/32E works best for our method and Ghard Top-2 while 100M/128E works better for Switch Transformer Top-1. Our method consistently outperforms the others across all the scales.

where some experts can receive a lot more tokens than the desired capacity. As a result, the step latency will be bottlenecked by the most loaded expert.

## 4.3 Scaling the Number of Experts

As presented in Table 1, increasing the number of experts effectively increases model capacity without increasing activated model size. We scale the number of experts while fixing the expert size to 100M parameters for both expert choice (EC) and GShard (Top-2) methods and find both methods work well in terms of perplexity on the evaluation dataset during pre-training. As demonstrated in Fig. 2 (b), having more experts consistently improves training perplexity.

## 4.4 Fine-tuning on GLUE and SuperGLUE

To validate whether improved perplexity directly translates to better performance in downstream tasks, we perform fine-tuning on 11 selected tasks from GLUE and SuperGLUE. We compare three MoE methods including Switch Transformer top-1 gating (ST Top-1), GShard top-2 gating (GS Top-2) and our method (EC-CF2) that matches the activation memory size and computational cost of GS Top-2. Indicated by the results in Table 2, our EC-CF2 method consistently outperforms the related methods and yields more than 2% average accuracy increase in a large 8B/64E setting. Table 3 further compares our 8B/64E model against its dense counterpart. Again, our method achieves stronger fine-tuning results, increasing the average score by 3.4 point.

Interestingly, we observe the 100M/32E model setting works the best for both GS Top-2 and EC-CF2, even though the effective model capacity is smaller than that of 100M/64E and 100M/128E. This result indicates that a good training perplexity does not always translate to better performance of downstream tasks.

| Model | BoolQ | CB | CoLA | MNLI | MRPC | QNLI | QQP | RTE | SST2 | WiC | WNLI | Avg |
|-------|-------|-----|------|------|------|------|------|------|------|------|------|------|
| Dense 8B | 88.2 | 100 | 86.4 | 91.3 | 86.7 | 94.7 | 91.2 | 92.2 | 97.2 | 75.6 | 78.1 | 89.2 |
| **EC-CF2 8B/64E** | 89.2 | 100 | 89.1 | 91.1 | 90.6 | 95.0 | 93.8 | 95.2 | 97.7 | 83.8 | 92.8 | **92.6** |

Table 3: Comparison between Dense 8B and Expert Choice (EC-CF2) 8B/64E models: Our method significantly outperforms the dense model in downstream tasks.

Table 4: (a) Limiting the number of experts per token in expert choice method affects downstream accuracy. (b) Comparing to Hash Layer.

| Method | Max # of Experts | Avg acc. |
|--------|------------------|----------|
| EC-CAP2 | 2 | $83.2 \pm 0.4$ |
| EC-CAP3 | 3 | $84.0 \pm 0.4$ |
| EC-CF2 | - | $84.0 \pm 0.2$ |
| Hash Layer | - | $81.3 \pm 0.1$ |

Figure 3: Distribution of the number of experts routed to per token in a 100M/64E model.

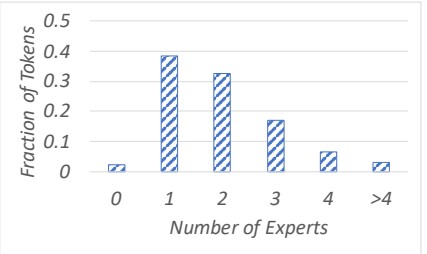

## 4.5 Heterogeneity Matters

**Capped Expert Choice:** We regularized expert choice by limiting the maximum number of experts for each token, using the method described in Section 3.3. Table 4 reports the average accuracy on the 11 selected datasets. EC-CAP2 is the variant of our expert choice method by limiting the number of experts of each token to 2. This decreases the fine-tuning accuracy by 0.8 points on average. In addition, EC-CAP3 allows a maximum of 3 experts per token and achieves on par results compared to the vanilla expert choice method. This ablation study confirms that **allowing variable number of experts per token is indeed helpful.**

**Variable Experts per Token:** We compute statistics on token-to-expert routing, particularly on the ratio of tokens that have been routed to a certain number of experts. According to Fig. 3, a majority of tokens have been routed to one or two experts while 23% have been routed to three or four experts and only about 3% tokens have been routed to more than 4 experts. This plot verifies our hypothesis that our method learns to allocate a variable number experts to tokens, which can be beneficial for important tokens.

## 4.6 Comparison with Hash Layer

In this section, we compare our method with Hash Layers [28]. We use $\mod x$ to map a token ID to an expert ID. This ensures load balance and generates specialized experts. The fine-tuning results are presented in the last row in Table 4. Hashing based routing performs worse than expert choice in terms of average scores and variance. **This indicates that load balancing alone does not generate all the benefits**.

## 4.7 Ablation

**Capacity Factor:** We study the capacity factor in our expert choice method and compare the training perplexity with the baseline top-1 gating method used in Switch Transformer. As described in Eq. (1), the capacity factor determines how many experts on average each token can be routed to, thus the bucket size $k$ of each expert. In all our previous experiments, we use a capacity factor of 2, which matches the computational footprint of the top-2 gating used in GShard method. To match the computation cost on a per-token basis fairly with top-1 gating used in Switch Transformer, we reduce the capacity factor to 1 and plot the training perplexity in Fig. 4 (a). Not surprisingly, using a smaller capacity factor yields higher perplexity, but our method still significantly outperforms top-1 gating. We further push the capacity factor down to 0.5, and observe that it still outperforms the top-1 gating.

**Comparison with Dense Models on Pre-training:** We compare our method with dense models on pre-training. As shown in Fig. 4 (b), our method consistently outperforms the dense method in

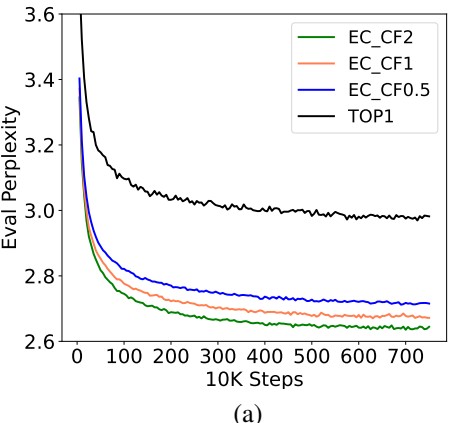 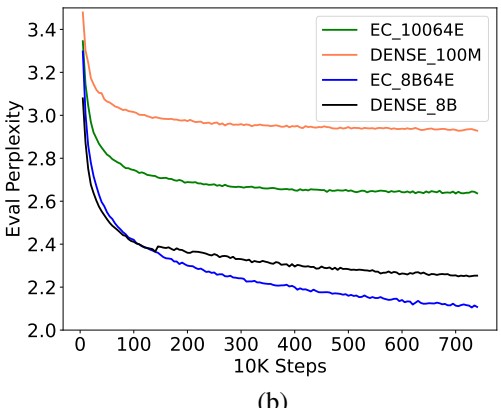

(a)                  (b)

Figure 4: (a) Varying the capacity factor in our expert choice method: Decreasing the capacity factor from two to one degrades the perplexity but still outperforms the top-1 gating. (b) Training perplexity comparison with dense models.

perplexity and convergence time. For a small expert size of 100M parameters, the benefit of sparse gating is even more significant. Orthogonal to results presented in Fig. 2 (b), where scaling the number of experts improves model performance, Fig. 4 (b) shows that increasing expert capacity also significantly increases model performance.

## 5 Conclusion

We propose a new routing method for sparsely activated mixture-of-experts (MoE) models. This method addresses load imbalance and under-utilization of experts in conventional MoE methods, and enables selecting different numbers of experts for each token. Our model demonstrates more than 2x training efficiency improvements when compared to the state-of-the-art GShard and Switch Transformer models, and also achieves strong gains when finetuning on 11 datasets in the GLUE and SuperGLUE benchmark.

## 6 Limitations

The expert choice method might not immediately apply to auto-regressive text generation as our current implementation takes in the past and future tokens to perform the top-$k$ selection. One possible solution is to collect a large batch of input sequences, dispatch tokens of the same sequence into separate groups, and perform expert choice routing for each group. Another scenario where the expert choice method does not immediately apply is when the batch size becomes very small during serving or inference. A global top-$k$ can be selected instead and we can cap the number of times each expert or token gets selected. We leave these possible improvements for future work.

Another long-standing issue with MoE has been the large memory footprint. Even though computational cost can be reduced using sparsely gated networks, the total number of parameters increases linearly or sub-linearly with the number of experts. Increasing the number of experts requires reservation of a large number of hardware devices. Therefore, dynamic (used) power is saved while static (reserved) power is not. Power saving techniques such as the ability to put hardware devices into low power states while not in use [17] can help with reducing the reserved power requirements.

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
