# A   Comparison on Fine-tuning with a Dense Model

Our 8B MoE model achieves stronger pre-training perplexity than its dense counterpart. However, a better perplexity does not always directly translate to downstream performance as demonstrated in Section 4.4. To this end, we compare fine-tuning performance of the 8B dense model and MoE model in Table 1. As shown in the table, our MoE model using expert choice routing consistently outperforms the dense model across the 11 tasks in GLUE and SuperGLUE.

| Model | BoolQ | CB | CoLA | MNLI | MRPC | QNLI | QQP | RTE | SST2 | WiC | WNLI | Avg |
|---|---|---|---|---|---|---|---|---|---|---|---|---|
| Dense 8B | 88.2 | 100 | 86.4 | 91.3 | 86.7 | 94.7 | 91.2 | 92.2 | 97.2 | 75.6 | 78.1 | 89.2 |
| **EC-CF2 8B/64E** | 89.2 | 100 | 89.1 | 91.1 | 90.6 | 95.0 | 93.8 | 95.2 | 97.7 | 83.8 | 92.8 | **92.6** |

Table 1: Comparison between Dense 8B and Expert Choice (EC-CF2) 8B/64E models: Our method significantly outperforms the dense model in downstream tasks.

# B   Capacity Factor

We evaluate the downstream task fine-tuning performance by varying the capacity factors. Note that a capacity factor of $n$ indicates on average how many experts each token can be received. EC-CF2 is our baseline expert choice, which matches GShard top-2 gating computational footprint. EC-CF1, however, matches Switch Transformer top-1 gating computational footprint. EC-CF0.5 further verifies that an aggressively lowered capacity factor can provide strong enough performance, that almost matches the top-2 gating baseline.

| Model | BoolQ | CB | CoLA | MNLI | MRPC | QNLI | QQP | RTE | SST2 | WiC | WNLI | Avg |
|---|---|---|---|---|---|---|---|---|---|---|---|---|
| Top-2 | 78.1 | 87.0 | 88.3 | 85.0 | 82.6 | 90.1 | 90.7 | 81.6 | 94.7 | 68.2 | 67.2 | 83.0±0.3 |
| EC-CAP2 | 78.2 | 88.0 | 88.5 | 85.7 | 83.0 | 90.8 | 91.1 | 80.0 | 95.4 | 70.4 | 64.1 | 83.2±0.4 |
| EC-CAP3 | 78.5 | 91.7 | 89.3 | 86.3 | 83.5 | 90.9 | 91.1 | 81.8 | 94.9 | 70.0 | 65.6 | 84.0±0.4 |
| EC-CF2 | 79.1 | 89.6 | 89.3 | 86.8 | 84.3 | 91.3 | 91.2 | 81.1 | 95.2 | 68.1 | 68.0 | 84.0±0.2 |
| EC-CF1 | 77.4 | 90.6 | 88.0 | 85.5 | 83.6 | 90.3 | 91.2 | 79.8 | 95.3 | 66.5 | 64.9 | 83.0±0.2 |
| EC-CF0.5 | 77.4 | 89.6 | 86.3 | 85.2 | 82.7 | 91.7 | 91.0 | 79.6 | 94.9 | 67.3 | 63.5 | 83.0 ±0.05 |
| Hash Layers | 76.1 | 85.2 | 86.7 | 83.4 | 82.5 | 90.0 | 90.3 | 75.7 | 94.0 | 67.4 | 63.3 | 81.3±1.0 |

Table 2: Comparison between different routing methods in fine-tuning of 100M/64E models. We perform 3 independent fine-tuning runs for each method and report the average results. This gives more accurate difference between the variants of expert choice method, since they achieve close fine-tuning results. We do not report averaged results in other experiments.

# C   Capped Expert Choice

As described in Section 4.5, the maximum number of experts each token is assigned can be capped by an entropy-regularized linear programming. Figure 1 compares the validation perplexity when training the 100M/64E models using the base expert choice method (EC-BASE), expert choice capped by two experts per token (EC-CAP2), expert choice capped by three experts per token (EC-CAP3), and GShard top-2 gating.

As shown in the figure, restricting the number of experts to 2 degrades the perplexity compared to the base expert choice method. This suggests that a more flexible allocation of experts (e.g. more than 2 experts for a token) can enhance model expressiveness. On the other hand, our EC-CAP2 and EC-CAP3 methods still outperform the top-2 gating method by a clear margin. We believe this confirms the effectiveness of a load balanced training, provided by our method. Finally, EC-CAP3 obtains comparable perplexity to EC-BASE. As indicated by Figure 3, only a little fraction of tokens use more than 3 experts therefore we see little or no difference between EC-BASE and EC-CAP3 variants. We present the fine-tuning results of these methods in Table 2.

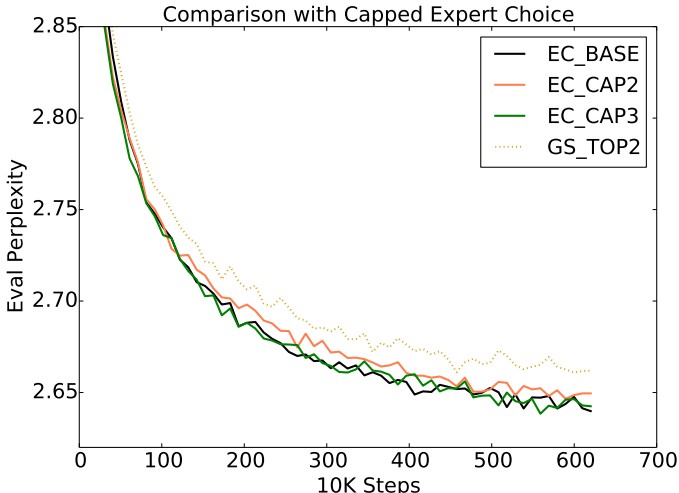

Figure 1: Validation perplexity during pre-training using various expert choice methods and top-2 gating.

## D Comparison with Hash Layer

In this section, we compare our method with Hash Layers [1]. We use  mod $x$ to map a token ID to an expert ID. This in some way ensures load balance and generates specialized experts. The fine-tuning results are presented in the last row in Table 2. Hashing based routing performs much worse than expert choice in terms of average scores and variance.

## E Fine-tuning Details

We did a hyperparameter search for both baseline models and expert choice method. For fine-tuning of the 8B dense model, we use a constant learning rate of 0.0001 and a dropout rate of 0.1. We freeze the attention layer and feed-forward layer while leaving the embedding and layer normalization trainable. This setting has been found optimal for the 8B dense model. For MoE 8B/64E models including GShard top-2 gating and expert choice, we found continuing the learning rate from the pre-trained model while using a square root learning rate decay works better. In addition, we do not apply parameter freezing for fine-tuning MoE models. For models with 100M expert size, we use a constant learning rate of 0.0001 and no dropout is used.

## References

[1] Stephen Roller, Sainbayar Sukhbaatar, Arthur Szlam, and Jason Weston. Hash layers for large sparse models, 2021.