# OpenReview forum: "Mixture-of-Experts with Expert Choice Routing"
_NeurIPS.cc/2022/Conference — NeurIPS 2022 Accept_

### Official Review · Reviewer_oGHv · 2022-06-19

**Rating:** 7
**Confidence:** 4
**Soundness:** 3 good
**Presentation:** 3 good
**Contribution:** 3 good

**Summary:**

The paper presents expert choice routing for mixture-of-experts models instead of the more common token-based routing strategies. The proposed method ensures ideal load balancing, resulting in faster training convergence as well as increased performance on downstream tasks, partly attributed through solving the under-spezialization problem in previous MOE approaches.

**Questions:**

- If I choose an appropriate hashing function for my token-based routing (cf. Balanced assignment [28]), shouldn't that ensure a balanced expert load and as such contradicts your statement that previous work "leads to an imbalanced load of experts" ?

**Limitations:**

Limitations are appropriately addressed.

**Strengths And Weaknesses:**

### Strengths:

- Simple but effective method.
- Expert choice routing demonstrates faster convergence and improved downstream performance.
- Insightful comparison towards Hashing Layers, confirming that just balancing the load has some performance drawbacks.
- Results are well presented and and claims are well supported by experimental results.

### Weaknesses:

- **(small)**: I think the paper would still benefit from a Figure similar to Figure 4 in [28] to drive home the point of load balancing.
- **(small)**: Adding a rough wall clock time for the training time would be helpful.

### Minor points:

- Next time make sure to omit $\verb+final+$ and $\verb+preprint+$ in the NeurIPS $\LaTeX$ package which adds line numbers and hence makes reviewing easier.
- Try out the `sort&compress` option to natbib which sorts the references and makes things a little cleaner presentation-wise.
- Clean up references to be consistent in terms of the included information, i.e conference citations should have the same style and make sure to handle preprints consistently and add their arXiv identifier. E.g. [2], [4], [18], [30] all have different formatting. [2] doesn't even contain the title.
- Table 1 Caption "Session 4.1" -> "Section 4.1". Generally, I found that you are inconsistent with adding cross reference links to quickly jump to Sections, e.g. in Section 4.5, the reference to Section 3.3 doesn't have one. Switch to `cleveref` and use $\verb|\Cref|$ for consistency and to avoid typos.
- Just below Table 2 there seems to be a lonely (b) with no corresponding (a) that most likely shouldn't be there.
- Table 2: There are a few errors in the printing of bold numbers and formatting. I always assume that bold number highlight the best performance, even if it's not the presented method. Hence, I'd suggest adding that as well as fixing the errors where one of the other methods actually performs better.  **100M/128E**: BoolQ, QNLI, QQP, WNLI -> ST or GS outperform reducing the outperformed tasks from 10/11 -> 6/11. Here also GS Top-2 should be printed bold for RTE. **100M/64E**: GS Top-2 should be bold for WiC. **100M/32E**: Can we get the number for ST Top-1 on CB for completeness? GS Top-2 should be bold for MRPC, RTE. **8B/64E**: GS Top-2 should be bold for BoolQ, MNLI, MRPC, SST2
- Appendix Table 2 EC-CF0.5 last column has additional space that shouldn't be there.

---

> ### Author Response · Authors · 2022-08-02
> **Response to Reviewer oGHv**
>
> We thank the reviewer’s detailed comments on formatting and results. We’ve updated the paper with:
> 1) Removing preprint from package import.
> 2) Added sort&compress option to natbib.
> 3) Replace \ref with \cref.
> 4) Fixing errors in Table 2.
> 5) Removing the additional space in the Appendix Table 2.
>
> **Q1: If I choose an appropriate hashing function for my token-based routing (cf. Balanced assignment [28]), shouldn't that ensure a balanced expert load and as such contradicts your statement that previous work "leads to an imbalanced load of experts"?**
>
> We thank the reviewer’s suggestion. We will change the statement to “Many of the prior work learns a token-to-expert mapping that can lead to an imbalanced load of experts.” We agree with the reviewer that hashing based routing can achieve near-perfect load balancing. Unlike hashing, our expert routing maintains a fixed expert capacity and makes sure the same number of tokens are routed to each experts. Therefore, bucket frequency plot like Figure 4 in [Hash Layer](https://arxiv.org/pdf/2106.04426.pdf) will be strictly a flat line in Expert Choice.

---

> > ### Comment · Reviewer_oGHv · 2022-08-02
> > **Thanks for the response**
> >
> > Yes, that's what I would've assumed for the load balancing plot. Regardless, I think often showing a side-by-side comparison of two routing methods where (a) shows an imbalance and (b) the proposed expert choice routing shows perfect balancing, is often helpful to catch the attention of a skimming reader and helps to visualize one of the main contributions. Anyways, this is just a suggestion and I believe the paper is already strong as is, even without it. Feel free to not add it if you don't see any value of having it.
> >
> > I still want to urge you to clean up the references for the final version as mentioned in the original review, there are quite a few of inconsistencies in papers cited from the same venue and consistency across venues is also just nice to see. This might seem nitpicky but it tremendously helps when trying to quickly parse the references.
> >
> > Great work, I really enjoyed the read!

---

> > > ### Author Response · Authors · 2022-08-03
> > > **Thanks for your response!**
> > >
> > > We cleaned up the reference and will provide a side-by-side comparison of two routing methods as suggested in the final version. Very appreciate your feedback.

---

### Official Review · Reviewer_1p81 · 2022-07-06

**Rating:** 7
**Confidence:** 5
**Soundness:** 4 excellent
**Presentation:** 3 good
**Contribution:** 4 excellent

**Summary:**

This work proposes a new routing method called Expert Choice Routing for the Mixture-of-Experts (MoE) architectures. In this method instead of routing each token to a specific number of (top-$k$) experts, each expert receives a specific number of (top-$k$) tokens. Naturally using this approach, as mentioned in the manuscript, the following properties will be held:
- Perfect load balancing will be achieved between all of the experts, since every expert receives the same number of tokens
- Different tokens can be routed to various number of experts

Similar to GShard [1] and Switch Transformer [2], the authors apply the MoE and gating logic in the dense feed forward layer (FFN). The authors also use these works as baselines to compare their work against.
This comparison shows that using the same amount of computation cost the Expert Choice Routing method achieves superior results on GLUE and SuperGLUE benchmarks. Additionally, it is shown that using the expert choice routing method, the training efficiency is almost doubled. Lastly, it is shown that when using expert choice routing, training perplexity consistently decreases as the number of experts are increased.

References

[1] Dmitry Lepikhin, HyoukJoong Lee, Yuanzhong Xu, Dehao Chen, Orhan Firat, Yanping Huang, Maxim Krikun, Noam Shazeer, and Zhifeng Chen. GShard: Scaling giant models with conditional computation and automatic sharding. In International Conference on Learning Representations, 2021.

[2] William Fedus, Barret Zoph, and Noam Shazeer. Switch transformers: Scaling to trillion parameter models with simple and efficient sparsity, 2021.

**Questions:**

- How different batch sizes affect the performance of expert choice (EC) routing? Are there regimes of batch sizes where using EC routing may not provide any advantages over token choice (TC) routing?
- What is the criteria for bolding numbers in Table 2? I assumed that highest number in each row is bolded. However, after inspecting the numbers, I found that this is not the case. For example, in the 100M/128 scenario on QQP, the highest number belongs to ST-Top-1 but the number belonging to EC-CF2 is highlighted.
- In Figure 2, it is clearly shown that EC routing improves the training efficiency compared to the GShard (GS). Was it also observed that eval perplexity at the end of the training is lower for EC? Does EC converge to a smaller eval perplexity compared to the GS at the end of the training?

**Limitations:**

The authors adequately addressed the limitations and potential negative impacts of this work in the manuscript.

**Strengths And Weaknesses:**

Strengths:
- The manuscript is well written. The method and background are explained clearly and are easy to follow.
- The paper introduces a novel routing technique which improves the training efficiency and accuracy of Mixture-of-Expert (MoE) architectures.
- The authors' central claims are sound, convincing, and well supported.

Weaknesses:
- There are few writing mistakes which should be fixed:
    - In Table 1, the 0.1B/128E model has 128 experts (not 64).
    - Incorrect citations (e.g. In [18] the paper name is missing. However, from pages 103 to 112 it can be inferred that the paper is GPipe)
    - In section 4.7, the term "We further pushes" should be changed to "We further push"
- In the last sentence of section 4.2, it is claimed that each GShard top-2 step is 20% slower than expert choice routing. However, this is not supported and explained anywhere within the manuscript. Additional clarification is necessary regarding this claim.

---

> ### Author Response · Authors · 2022-08-02
> **Response to Reviewer 1p81**
>
> We thank the reviewer's detailed feedback on formatting and grammar. We fixed the errors pointed out in the updated version, highlighted in red. We would like to address the questions one by one:
>
> **Q1: Why is GShard top-2 step 20% slower than expert choice routing?**
>
> We discussed the reasons for expert choice being faster in step time in Section 3.1 “Load Imbalance”: Load imbalance can also affect step latency, thus inference time, as the step latency can be determined by the most loaded expert. Token-based routing requires over provisioning expert buffer size to mitigate the effect of dropping tokens. In the 8B64E setting, the over capacity factor can be as large as eight empirically in a traditional GShard top-2 method. This leads to GShard top-2 being 20% slower in step time than expert choice routing. We will add more explanation in Section 4.2 as highlighted in red in the updated version.
>
> **Q2: How does batch size affect EC performance?**
>
> We empirically find that varying the batch size (e.g. reducing from 1024 to 512) but fixing the total pretraining tokens won’t affect model performance. We would like to add more comparison and analysis in the final version.
>
> **Q3: What is the criteria for bolding numbers in Table 2?**
>
> The reviewer is correct the highest number in each row should be bolded. We fixed the bolding numbers in Table 2 in the updated version.
>
> **Q4: Was it also observed that eval perplexity at the end of the training is lower for EC? Does EC converge to a smaller eval perplexity compared to the GS at the end of the training?**
>
> Yes, the evaluation perplexity at the end of training (with fixed total training tokens) is also lower in EC compared to GShard Top-2.

---

> > ### Comment · Reviewer_1p81 · 2022-08-08
> > **Thanks for the Response and Modifications**
> >
> > I would like to thank the authors for addressing the issues and also providing clear answers. I can confirm that my questions are answered and also the issues that I brought up with both manuscript and references are now resolved.

---

### Official Review · Reviewer_HqBR · 2022-07-09

**Rating:** 6
**Confidence:** 5
**Soundness:** 3 good
**Presentation:** 3 good
**Contribution:** 3 good

**Summary:**

The paper works on sparse MoE for large-scale Transformer.

Key idea: Rather than letting tokens select the top-k experts, the proposed method selects the top-k tokens for each expert.



**Questions:**

- How about the finetuning numbers of dense models in Table 2?

- "model training follows the setups of GLaM": is the model pretrained using the GPT task?

- The results are compared with the T5 dense model, but the models use different training data. The numbers are not fairly comparable. How about the results using the same data?

- Are there numbers of training the model on public text data, such as CommonCrawl?

- How about the results for vision Transformers or machine translation instead of just language modeling?

**Limitations:**

There are two limitations mentioned in Sec 6.

- The proposed method can not immediately apply to auto-regressive text generation.

- Large memory footprint.

**Strengths And Weaknesses:**

Strengths:

1. The idea of selecting tokens for experts to experts is novel for MoE model training.

2. Better convergence speed compared with previous methods.

3. The method naturally solves the load balance issue of MoE.


********************************************
Weaknesses:

1. It is non-trivial to apply the proposed method to the autoregressive generation of GPT-like models, because all the tokens are not pre-given at the beginning. However, the most successful MoE-based models are GPT applications, which restricts the usage of the proposed method. This limitation is also mentioned in Sec 6.

2. The fine-tuning performance of dense models is not reported in Table 2. Only MoE-based models are evaluated under the transfer setting. It is straightforward to compare the finetuning numbers for dense models.

3. The results are compared with the T5 dense model, but the models use different training data. The numbers are not fairly comparable.

4. The training data are not publicly available. It makes future comparisons difficult for other papers.

---

> ### Author Response · Authors · 2022-08-02
> **Response to Reviewer HqBR**
>
> **Q1: How about the finetuning numbers of dense models in Table 2? & The results are compared with the T5 dense models but the models use different training data.**
>
> Thanks the reviewer pointing out a confusion in the paper. In our supplementary section A, we provided the finetuning results of a 8B dense model trained using the same data as our sparse models. Our best sparse model outperforms the 8B dense model by 3.4 points on average. We agree that our results are not directly comparable to those of T5 due to the use of different data. We removed the statement comparing with T5 and moved the detailed finetunine comparison with a 8B dense model to the main Section 4.4 highlighted in red. Thank you for raising the question!
>
> **Q2: Model training follows the setups of GLaM: is the model pretrained using the GPT task?**
>
> No, we do not use GPT few-shot tasks. Pre-training is down-stream task agnostic.
>
> **Q3: Are there numbers of training the model on public text data, such as CommonCrawl?**
>
> Our data contains a filtered version of webpages from CommonCrawl and includes other data such as Wikipedia. We follow the same dataset creation procedure described in Section 3 of [GLaM](https://arxiv.org/pdf/2112.06905.pdf).
>
> **Q4: How about the results for vision Transformers or machine translation instead of just language modeling?**
>
> Yes, for the main results presented in the paper, we chose primarily sequence classification tasks in GLUE and SuperGLUE. We add additional evaluations on generative tasks for the final paper. With proper treatment on expert capacity, we demonstrate strong **zero-shot** results on WMT and **one-shot** results on TriviaQA. In the presented results in the table, **we only use 100B pretraining tokens for the limited time while GPT3 used 300B pretraining tokens. Our 1B64E model activation memory (FLOPS) is close to GPT3 1.3B** due to the interleaved dense and MoE layers in our model as explained in Section 4.1. EC generates comparable results as [GLaM](https://arxiv.org/pdf/2112.06905.pdf) or [GShard](https://arxiv.org/abs/2006.16668) Top-2 gating, while being stronger in Ro->En, En->Fr, En->De, En->Ro. We thank the reviewer's insightful feedback on GPT-style generative tasks and would like to incorporate **zero-shot and one-shot results on WMT and TriviaQA** in the final version.
>
> |  Model          | Fr->En 14 | En->Fr 14 | De->En 16 | En->De 16 | Ro->En 16 | En->Ro 16 |  TriviaQA One-Shot|
> | ----------- | ----------- | ----------- | ----------- | ----------- | ----------- | ----------- | ----------- |
> | Top-2 1B64E |    **25.1**   |     5.84     |   **23.8**    |    5.69        |      21.8      |      3.60        |  42.4 |
> | **EC 1B64E**  |      24.8      |     10.7     |    23.1         |     8.20      |    **23.2**  |      **4.70**   | **42.5** |
> | GPT3  1.3B    |      3.60       |     2.82     |    4.04        |    2.92        |      3.56      |      3.09       |  26.5 |
> | GPT3  2.7B     |      21.2     |   **19.3**  |   22.5        |     **13.7**      |      16.8      |     4.26    |  35.9  |

---

> > ### Comment · Reviewer_HqBR · 2022-08-05
> > **table**
> >
> > Could you re-format the table in the response by using markdown? It's difficult to read the numbers now. Thank you!
> >
> > | Model | Fr->En 14 | En->Fr 14 | De->En 16 | En->De 16 | Ro->En 16 | En->Ro 16 | TriviaQA One-Shot|
> >
> > | Top-2 1B64E | 25.1 | 5.84 | 23.8 | 5.69 | 21.8 | 3.60 | 42.4 |
> >
> > | EC 1B64E | 24.8 | 10.7 | 23.1 | 8.20 | 23.2 | 4.70 | 42.5 |
> >
> > | GPT3 1.3B | 3.60 | 2.82 | 4.04 | 2.92 | 3.56 | 3.09 | 26.5 |
> >
> > | GPT3 2.7B | 21.2 | 19.3 | 22.5 | 13.7 | 16.8 | 4.26 | 35.9 |

---

> > > ### Author Response · Authors · 2022-08-05
> > > **Fixed the table.**
> > >
> > > Thank you for the suggestion. We've fixed the table using markdown.

---

### Official Review · Reviewer_49ur · 2022-07-11

**Rating:** 7
**Confidence:** 5
**Soundness:** 3 good
**Presentation:** 4 excellent
**Contribution:** 4 excellent

**Summary:**

The paper proposes a new token routing strategy for the MoE model architecture. The existing methods mostly focus on how each token chooses experts to be processed by, but this paper designs the opposite way which each expert chooses which tokens to process. By doing this, the authors claim that MoE models' intrinsic load balancing issue is resolved naturally. Furthermore, by applying more computation to more important tokens, the paper presents faster convergence and higher performance on a GLUE and SuperGLUE tasks.

**Questions:**

1. In equation (2), can you clarify which dimension is used to do softmax?
2. Can you clarify why subsets of tasks from GLUE and SuperGLUE are selected? Are they all sequence classification tasks? If that's the case, I wondered how expert choice works for token classification tasks (ReCORD). Some of less importance tokens might not be fully processed and could potentially cause inaccurate perditions for them.
3. When experts choose tokens, do they make choices across different sentences in a batch? If so, how would it affect the training dynamics? Can a model prefer some specific sentences? If not, could you explain how a batch is handled?

**Strengths And Weaknesses:**

MoE models are getting more widely used due to their efficient scaling property with higher quality than the vanilla dense models. However, there has been a load balancing issue from the beginning, and it is known to be critical to the model performance. Still, it is hard to solve and prevents models to be effectively trained such as under-trained experts. Given this situation, the paper proposes a very unique and noble approach to solve the problem naturally. This is very timely and makes sense intuitively. The paper is well-organized, clearly written and easy to follow.

---

> ### Author Response · Authors · 2022-08-02
> **Response to Reviewer 49ur**
>
> **Q1: In equation (2), can you clarify which dimension is used to do softmax?**
>
> We apply softmax normalization over the expert dimension while the top-k selection (for each expert) is done along the token dimension.
>
> **Q2: Some less important tokens might not be fully processed. How tasks are selected?**
>
> In our method, each transformer block has an independent gating network thus even tokens can be dropped in some layers, they will be processed in other layers. Also, in our proposed model architecture, dense layers and sparse layers are interleaved, therefore, dense layers ensure the process of all tokens. In addition, token-based routing can also drop tokens due to load imbalance, as explained bolded texts in Section 3.1. Our method mitigates dropping tokens with perfect load balance.
>
> Yes, for the main results presented in the paper, we chose primarily sequence classification tasks in GLUE and SuperGLUE. We add additional evaluations on generative tasks for the final paper. With proper treatment on expert capacity, we demonstrate strong **zero-shot** results on WMT and **one-shot** results on TriviaQA. In the presented results in the table, **we only use 100B pretraining tokens for the limited time while GPT3 used 300B pretraining tokens. Our 1B64E model activation memory (FLOPS) is close to GPT3 1.3B** due to the interleaved dense and MoE layers in our model as explained in Section 4.1. EC generates comparable results as [GLaM](https://arxiv.org/pdf/2112.06905.pdf) or [GShard](https://arxiv.org/abs/2006.16668) Top-2 gating, while being stronger in Ro->En, En->Fr, En->De, En->Ro. We thank the reviewer's insightful feedback on benchmarking and would like to incorporate ReCORD finetuning, zero-shot and one-shot results on WMT and TriviaQA in the final version.
>
> |  Model          | Fr->En 14 | En->Fr 14 | De->En 16 | En->De 16 | Ro->En 16 | En->Ro 16 |  TriviaQA One-Shot|
> | ----------- | ----------- | ----------- | ----------- | ----------- | ----------- | ----------- | ----------- |
> | Top-2 1B64E |    **25.1**   |     5.84     |   **23.8**    |    5.69        |      21.8      |      3.60        |  42.4 |
> | **EC 1B64E**  |      24.8      |     10.7     |    23.1         |     8.20      |    **23.2**  |      **4.70**   | **42.5** |
> | GPT3  1.3B    |      3.60       |     2.82     |    4.04        |    2.92        |      3.56      |      3.09       |  26.5 |
> | GPT3  2.7B     |      21.2     |   **19.3**  |   22.5        |     **13.7**      |      16.8      |     4.26    |  35.9  |
>
> **Q3: When experts choose tokens, do they make choices across different sentences in a batch?**
>
> Yes, expert choice makes routing decisions across multiple sentences in a batch. It is possible that certain sentences use more FLOPs and more experts. In our analysis section, we show that having variable compute per token is indeed beneficial. We will provide additional analyses such as the routing dynamics at the sentence level in our next version. Thank you for raising this great question!

---

> > ### Comment · Reviewer_49ur · 2022-08-08
> > **Thanks for the response!**
> >
> > The answers clarified my questions.
> > I will keep my rating Accept for the current revised version.
> >
> > I have just one follow-up question which doesn't necessarily need to be updated in the paper. In some of my experiments, I observed softmax on token dimension worked better. Have you done any experiments with that?

---

### Author Response · Authors · 2022-08-02
**Thank you letter**

Many thanks to the anonymous reviewers for their dedication in the review process. Their professional comments and questions made us realize that we have much to improve. We addressed many detailed questions in the updated version, highlighted in red.

---

### Meta-Review · Area_Chair_bh4w · 2022-08-27

**Recommendation:** Accept
**Confidence:** Certain

**Metareview:**

This work introduces a new token routing strategy for MoE models.  Instead of allocating a fixed number of experts to each token using a top-k function regardless of the relative importance of different tokens, the proposed strategy adopts a heterogeneous mixture-of-experts employing an expert choice method.

The proposed method is simple, training efficient, and empirically effective. The paper is well-written and easy to follow. Overall, it is a good paper.


**Award:**

No

---

### Decision · Program_Chairs · 2022-09-14

Accept